# First-Principles Study of Nitrogen Adsorption and Dissociation on ZrMnFe(110) Surface

**DOI:** 10.3390/ma16093323

**Published:** 2023-04-24

**Authors:** Qiaobin Yang, Fanhao Zeng, Meiyan Chen, Yu Dai, Yafang Gao, Rui Huang, Yi Gu, Jiangfeng Song

**Affiliations:** 1Powder Metallurgy Research Institute, Central South University, Changsha 410083, China; 2College of Materials Science and Engineering, Central South University, Changsha 410083, China; 3China Academy of Engineering Physic, Mianyang 621900, China; iterchina@163.com

**Keywords:** density functional theory, adsorption, dissociation, reaction pathway

## Abstract

The adsorption, dissociation and penetration processes of N_2_ on the surface of ZrMnFe(110) were investigated using the first-principles calculation method in this paper. The results indicate that the vacancy Hollow 1 composed of 4Zr1Fe on the surface of ZrMnFe(110) is the best adsorption site for the N_2_ molecule and N atom, and the adsorption energies are 10.215 eV and 6.057 eV, respectively. Electron structure analysis indicates that the N_2_ molecule and N atoms adsorbed mainly interact with Zr atoms on the surface. The transition state calculation shows that the maximum energy barriers to be overcome for the N_2_ molecule and N atom on the ZrMnFe(110) surface were 1.129 eV and 0.766 eV, respectively. This study provides fundamental insight into the nitriding mechanism of nitrogen molecules in ZrMnFe.

## 1. Introduction

The development of getter materials for gas capture and storage is of great significance to clean energy applications [1,2], gas purification and separation [3] as well as the recovery and utilization of nuclear reactor exhaust [4,5,6,7,8,9]. Getter materials used for gas capture include metal-organic frame materials, carbon-based materials, polymers, alloy getter, etc. [7,8,9,10,11]. These kinds of getter materials are special functional materials which can effectively adsorb some active gases (H_2_, CO, N_2_, O_2_, CO_2_, H_2_O and Cm, Hn) through physical or chemical reactions [12,13,14,15]. Among them, ZrMnFe alloy containing Laves single-phase compound plays a major role in the treatment of foreign gas [16,17,18,19,20,21,22].

There has been some work on the inspiratory performance of ZrMnFe [23,24,25]. Klein et al. [4] tested the decomposition performance of methane under N_2_ atmosphere in the laboratory and discovered that N_2_ could inhibit the decomposition performance of ST909 to methane. James et al. [18] tested and compared the decomposition performance and ZrMnFeAl, AlNiFe and ZrNi for CH_4_ and NH_3_. The results showed that the decomposition performance of CH_4_ was affected by preparation method. According to our previous work [26], the nitriding process of ZrMnFe can be described as follows: Nitrogen molecules are absorbed at the surface and dissociated into nitrogen atoms at the surface, and then nitrogen atoms diffuse deeper into the matrix, forming new ZrN phases, and eventually ZrN particles grow into a barrier layer, further preventing the penetration of nitrogen. The adsorption and activation of N_2_ is the initial step of ZrMnFe getter. As a result of N_2_ in ZrMnFe getter, adsorption on the surface of the reconciliation process is very complex and surface adsorption reaction is difficult to observe through the experiment, with little understanding of the reaction mechanism. On the other hand, theoretical calculations can help simulation and prediction to provide useful information for experiments. However, there is little research on its inspiratory mechanism by first-principles calculation method so far. Based on density functional theory, this paper studies the adsorption, dissociation and infiltration processes of the N_2_ molecule on the ZrMnFe(110) surface by using the plane wave pseudopotential method. In particular, we analyze energy and electronic structures in the nitriding process. Finally, the lowest energy paths of separation and infiltration are calculated. The influence mechanism of the microstructure and surface structure of ZrMnFe alloy on the adsorption properties of N_2_ is comprehensively studied, which provides a theoretical basis for the preparation of a new generation of inspiratory alloy.

## 2. Calculation Model and Method

ZrMnFe is one of the representatives of Laves phase impurity removal alloy. The space group is P6/mmc, and cell parameters are a = 5.045 Å, b = 4.856 Å, c = 8.029 Å, α = β = 90° and γ = 120° [27]. The cell model is shown in Figure 1a. The cell parameters of the optimized structure were a = 4.943 Å, b = 4.720 Å and c = 8.251 Å, which were in good agreement with the experimental values. On this basis, the surface structure of ZrMnFe(110) was obtained by cutting ZrMnFe alloy. The ideal surface structure for ZrMnFe(110) cleaning is shown in Figure 1b. The surface structure of six atomic layers was adopted in the calculation, and the height of vacuum layer was set as 15 Å. The coordinates of the bottom four layers were fixed and the rest of the atoms were allowed to relax in the calculation process.

The adsorption of N_2_ molecules and N atoms on the ZrMnFe(110) surface was simulated by using the pseudopotential plane wave method based on first principles in this paper [28,29,30]. Firstly, a density mixing scheme combined with the conjugate gradient method of LBFGS (large Broyden–Fletcher–Goldfarb–Shanno) [31,32,33,34] was used to optimize the geometric structure of the constructed system and obtain a stable structure. Then, the energy of various adsorption sites was calculated, respectively, and the adsorption energy, electron state density and differential charge density of the optimized system were calculated, respectively. The valence electron wave function of the system was expanded by plane wave base vector, with the truncation energy of the plane wave assumed to be 500 eV. The iterative convergence accuracy was 2 × 10^−6^ eV. The PBE functional under the generalized gradient approximation (GGA) is used to describe exchange correlation energy [35,36,37], and the interaction between ion and electron was calculated by using ultra-soft pseudo potential. The total energy was calculated in the reciprocal space. The Brillouin region integral is adopted by the Monkhorst–Pack method, with K point being taken as 3 × 3 × 1 [38]. The geometric optimization, adsorption energy and electronic structure were calculated by the CASTEP quantum module and the transition state by the DMol3 module.

## 3. Calculation Results and Discussion

The adsorption of the N atom and N_2_ molecule on the ZrMnFe(110) surface was systematically discussed, as shown in Figure 2. It can be seen from the figure that there are five adsorption sites with high symmetry potential: (i) Top 1 site is right above the Zr atom on the surface; (ii) Top 2 site is directly above the Fe atom on the surface; (iii) The Bridge site is the midpoint of a surface-adjacent Zr atom; (iv) Hollow 1 site is the center of a pentagon surrounded by four Zr atoms and one Fe atom and (v) Hollow 2 site is the center of a triangle surrounded by one Zr atom and two Fe atoms.

Adsorption energy Ea is the difference in the total energy of the system before and after adsorption, which is calculated by [39]:(1)$$E_{a}=E_{ZrMnFe}+E_{N}-E_{ZrMnFe/N}$$
(2)$$E_{m}=E_{ZrMnFe}+E_{N_{2}}-E_{ZrMnFe/N_{2}}$$
where $E_{a}$, $E_{m}$, $E_{ZrMnFe}$, $E_{N}$, $E_{N_{2}}$, $E_{ZrMnFe/N}$, and $E_{ZrMnFe/N_{2}}$, respectively, represent the adsorption energy, the energy of the clean ZrMnFe(110) surface, the energy of N atoms, the energy of N_2_ molecules, the total energy of the N atom adsorbed on the ZrMnFe(110) surface and the total energy of N_2_ molecule adsorbed on the ZrMnFe(110) surface.

### 3.1. Adsorption of N Atoms on ZrMnFe(110) Surface

The five adsorption configurations of N atoms on the surface of ZrMnFe(110) were optimized. The structural changes before and after the adsorption of N atoms are shown in Figure 3. As can be seen from the figure, there were only stable adsorption sites for N atoms, namely, Hollow 1 and Hollow 2, while all the N atoms at Top 1, Top 2 and Bridge sites moved to the hollow site.

In combination with the adsorption energy of N atoms at each position on the surface of ZrMnFe(110) and the distance from the surface in Table 1, the vacancy Hollow 1 composed of 4Zr1Fe is the best adsorption site for the N atom on the ZrMnFe(110) surface, and the adsorption energy of N atom is 6.057 eV. At the same time, the adsorption energy of the N atom on the vacancy Hollow 2 composed of 1Zr2Fe is 5.592 eV. In addition, the distance between the N atom adsorbed at Hollow 1 site and the top surface of ZrMnFe(110) was 0.659 Å, while that between the N atom adsorbed at Hollow 2 site and the top surface of ZrMnFe(110) was 3.357 Å, indicating that the N atom is more likely to be adsorbed at Hollow 1 site, which is also conducive to further reactions of N atoms at this site.

### 3.2. Adsorption of N_2_ Molecule on ZrMnFe(110) Surface

In order to study the adsorption of nitrogen molecules (N_2_) on the surface of ZrMnFe(110) in detail, five highly symmetric adsorption sites were selected, as shown in Figure 2. Two different adsorption configurations were obtained for each adsorption site according to the orientation of the diatomic molecule. Taking the N_2_ molecule at Hollow 1 site as an example, the adsorption configuration was denoted as “H1-Hor” when the N_2_ molecule was placed horizontally on the surface of ZrMnFe(110) and as “H1-Ver” when the N_2_ molecule was perpendicular to the surface of ZrMnFe(110). Figure 4 displays ten adsorption sites initially set by the N_2_ molecule on the surface of ZrMnFe(110).

The structural changes after ten adsorption configurations of the N_2_ molecule on the surface of ZrMnFe(110) were optimized and are shown in Figure 5. The results showed that the two types of N_2_ molecules adsorbed at the B site (Hor-B and Ver-B) moved to the H1-Hor site and remained unchanged at the H1-Hor site after optimization, while the position and morphology of N_2_ molecules at other adsorption sites changed to some extent. The calculated adsorption energy and parameters are shown in Table 2. It can be seen from the table that Hollow 1 is the best adsorption site for the N_2_ molecule on the surface of ZrMnFe(110) after stable adsorption of the N_2_ molecule. The optimal adsorption energy is 10.215 eV, and the orientation of the N_2_ molecule is horizontal.

In addition, the N_2_ molecule is not easy to dissociate due to its strong triple bond. Even so, when the N_2_ molecule is adsorbed on the surface of ZrMnFe(110), the strength of the triple bond of N and N will be weakened. As can be seen from Table 2, the N-N distance relaxed from 1.099 to 1.350 Å at the Hollow 1 site, and relaxed to some extent at other adsorption sites, which indicates that Hollow 1 was the optimal adsorption site for the N_2_ molecule on the surface of ZrMnFe(110).

### 3.3. Analysis of Electronic Structure

The results of adsorption energy calculation showed that Hollow 1 site on the surface of ZrMnFe(110) was the optimal adsorption site for both the N_2_ molecule and N atom. In order to further analyze the influence of the N_2_ molecule and N atom on the interactions between surface atoms, the electronic structures of the N atom and N_2_ molecule adsorbed on the ZrMnFe(110) surface were calculated, respectively. By observing the charge transfer between atoms, the differential charge density revealed the bonding properties between atoms at the adsorption site.

Figure 6 shows the differential charge density diagram of the N_2_ molecule and N atom adsorbed at Hollow 1 site on the surface of ZrMnFe(110), where the blue area represents electron loss and the red area represents electron gain. It can be seen from Figure 6a that the N_2_ molecule gained electrons, while the Zr atom lost electrons, accompanied by a large amount of overlapping in the electron clouds between the two N atoms, indicating the existence of a N-N bond. However, there were fewer overlapping areas in the electron clouds between the N atom and adjacent Zr atom, indicating that an ionic N-Zr bond had been formed, but was weaker than the N-N bond. As can be seen from the differential charge density diagram adsorbed by the N atom in Figure 6b, the number of overlapping electron clouds between the N atom and adjacent Zr was significantly higher than that in Figure 6a, indicating that the interaction between the N atom and Zr atom was enhanced after the dissociation of the N_2_ molecule. More importantly, the electron cloud of the N atom was not uniformly distributed around it but towards the side of the Zr atom, which indicates that N-Zr bond also had covalence.

Figure 7 compares the DOS diagrams before and after adsorption of the N atom at H1 site on the ZrMnFe(110) surface. It can be seen from the figure that the shape of the total state density diagram on the clean and adsorbed surfaces was basically the same and the total state density of the adsorbed surface newly peaked at −14.05 eV. Figure 7c shows the PDOS of the N atoms and Zr atoms before and after adsorption. After adsorption, new density peaks emerged at −14.05 eV on the 5s, 4p and 4d orbitals of Zr, which overlapped and resonated with those in the 2s orbitals of the N atoms, indicating orbital hybridization and proving the formation of N-Zr covalent bonds.

Figure 8 compares the DOS diagrams before and after the adsorption of the N_2_ molecule at Hollow 1 site on the ZrMnFe(110) surface. As can be seen from the figure, the shape of the total state density diagram basically remained unchanged after adsorption, but the total state density of the adsorption surface formed newly appeared peaks at −20.61 eV and −11.93 eV, respectively. It can be seen from PDOS that these two peaks were contributed by the state density peak of the N atom, indicating that the state density of the N_2_ molecule were split after adsorption. Hence, it can be concluded that the N_2_ molecule had been dissociated to a certain extent. In addition, as shown in Figure 8c, new density peaks appeared at −20.61 eV and −11.93 eV in the 4p orbitals of Zr after adsorption, respectively, which was also attributed to the hybridization of the N atom by the Zr atomic orbital.

### 3.4. Interface Reaction

Dissociation chemisorption is a key step in most surface chemistry. Therefore, in order to study the nitridation mechanism of the ZrMnFe(110) surface, the adsorption and dissociation characteristics of N_2_ molecules should be studied in detail.

As mentioned above, N_2_ molecules would preferentially adsorb on the H1-Hor site on the surface of ZrMnFe(110). After adsorption, the N-N bond became longer but weaker. In the follow-up study, we expected the nitrogen molecule (N_2_) to adsorb on the surface of ZrMnFe(110) and dissociate. Therefore, the transition state of the N_2_ molecular dissociation process was studied by combining linear cooperative transformation (LST) with quadratic cooperative transformation (QST) and conjugate gradient (CG) [40,41,42]. The dissociation process under the minimum energy barrier was determined by calculating reaction pathways.

Figure 9 shows the initial, final and intermediate states of the dissociation of N_2_ molecules on the ZrMnFe(110) surface. The energy of the final configuration in the dissociation process was found to be lower than that of the initial configuration, indicating that the dissociation reaction is easily able to proceed both thermodynamically and kinetically. The difference in capacity between the initial and final configurations was about 0.418 eV. In addition, it can be seen from the figure that a maximum of 1.129 eV energy needs to be surpassed during dissociation reaction, indicating that it is difficult for N_2_ molecules to dissociate. The specific process is described as follows: In the initial configuration, the N_2_ molecule was statically adsorbed on vacancy Hollow 1. With the occurrence of dissociation reaction, the horizontally oriented N_2_ molecule gradually tilted and the up-facing N atom in the N_2_ molecule gradually moved to other vacancies on the ZrMnFe(110) surface until it moved to the vacancy. Two N atoms were adsorbed on the two adjacent vacancy Hollow 1, eventually forming a co-adsorption state.

Based on the above method, we calculated the transition state of the N atom infiltrating from the ZrMnFe(110) surface to the inner layer. The optimal path of the N atom penetration is shown in Figure 10. The results demonstrated that the maximum energy barrier to be overcome during the N atom penetration was 0.766 eV, lower than that in the case of N_2_ molecule dissociation.

## 4. Conclusions

(1)The adsorption behavior of the N atom and N_2_ molecule on the ZrMnFe(110) surface was investigated theoretically by using the first-principles calculation method. The results show that vacancies Hollow 1 and H1-Horis are the most stable adsorption configurations of the N atom and N_2_ molecule on the ZrMnFe(110) surface, with adsorption energy of 6.057 eV and 10.215 eV, respectively. The N_2_ molecular bond is activated after adsorption, with its length relaxing from 1.099 to 1.350 Å.(2)Differential charge density analysis shows that both N atoms and N_2_ molecules gain electrons after adsorption, while Zr atoms on the ZrMnFe(110) surface lose electrons, forming a N-Zr ionic bond, but N_2_ molecules do not completely dissociate with the N-N bond, which is attributed to the hybridization of the 5s, 4p and 4d orbitals of Zr and the 2s orbitals of N after adsorption according to the density analysis of the bound states.(3)The calculated transition states show that the maximum energy barrier to be overcome during the dissociation of the N_2_ molecule on the surface of ZrMnFe(110) is 1.129 eV, while that during the permeation of the N atom from the surface of ZrMnFe(110) is lower at 0.766 eV. Moreover, N_2_ molecules are more likely to undergo osmotic reaction after dissociation reaction on the ZrMnFe(110) surface. All these results can provide theoretical guidance for exploring the adsorption mechanism of the N_2_ molecule on the ZrMnFe(110) surface.

## Figures and Tables

**Figure 1 materials-16-03323-f001:**
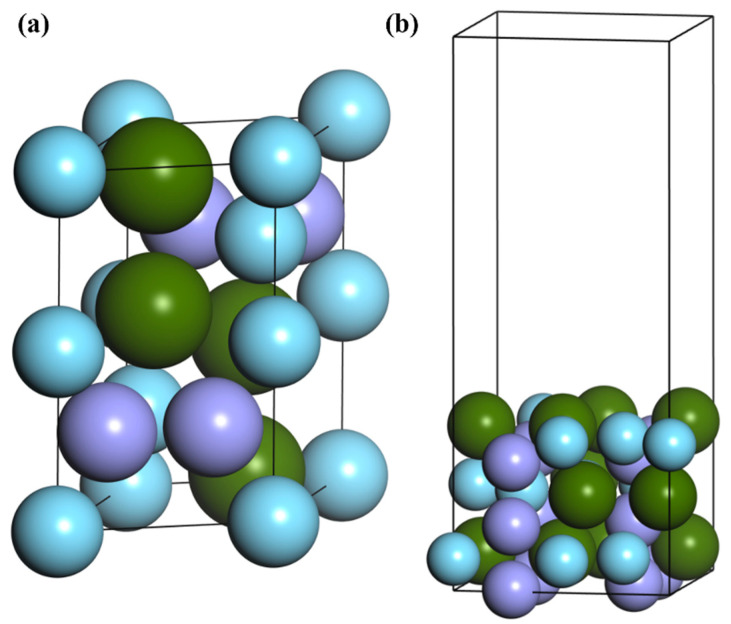
(**a**) ZrMnFe cell model and (**b**) crystal model of clean ZrMnFe(110) surface.

**Figure 2 materials-16-03323-f002:**
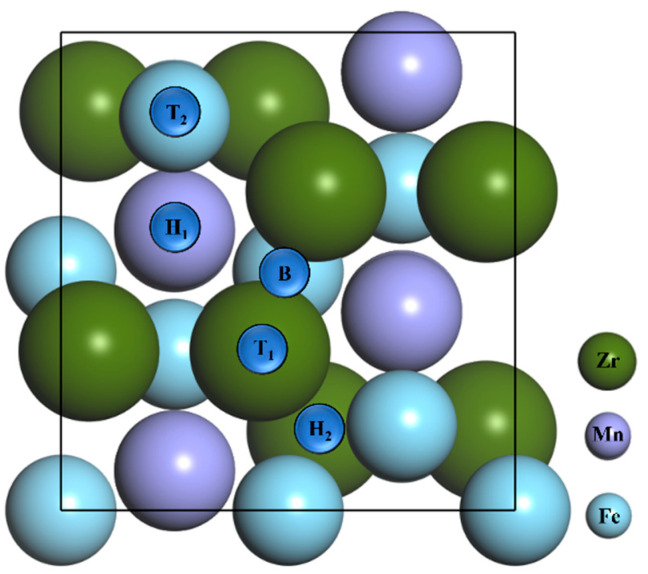
Top view of ZrMnFe(110) clean surface model and each adsorption site, where T1 = Top 1 site, T2 = Top 2 site, H1 = Hollow 1 site, H2 = Hollow 2 site and B = Bridge site.

**Figure 3 materials-16-03323-f003:**
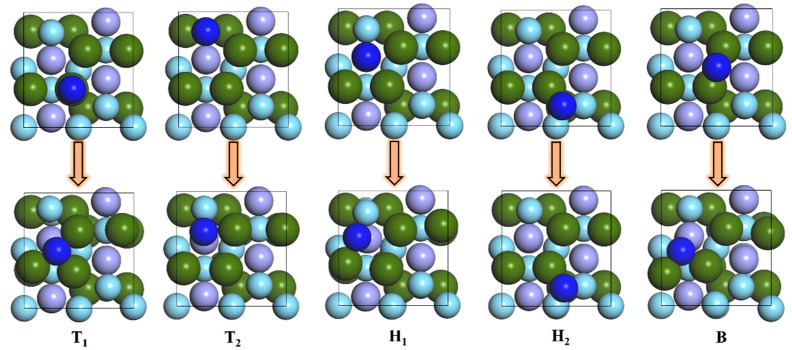
Changes of N atomic adsorption sites on ZrMnFe(110) surface after optimization.

**Figure 4 materials-16-03323-f004:**
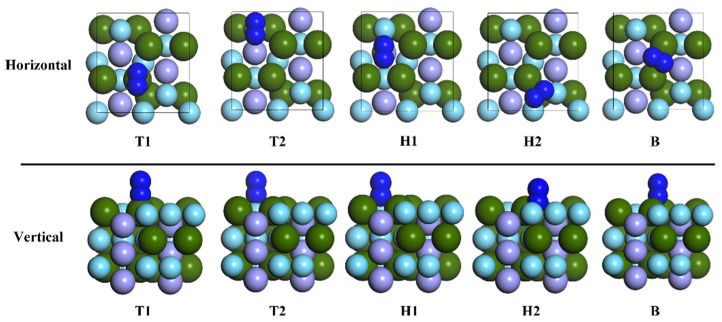
Adsorption sites of N_2_ molecule on the surface of ZrMnFe(110).

**Figure 5 materials-16-03323-f005:**
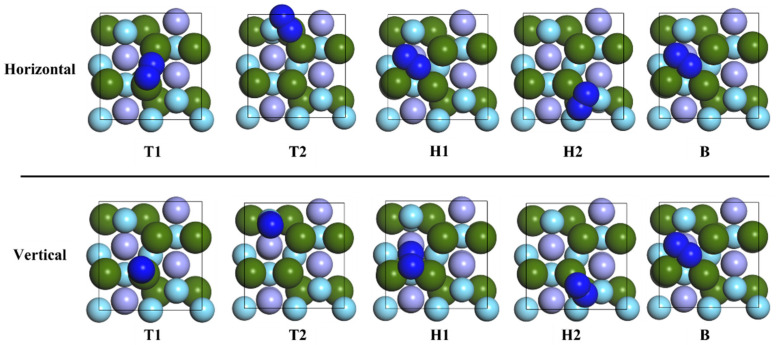
The adsorption site of N_2_ molecule on ZrMnFe(110) surface.

**Figure 6 materials-16-03323-f006:**
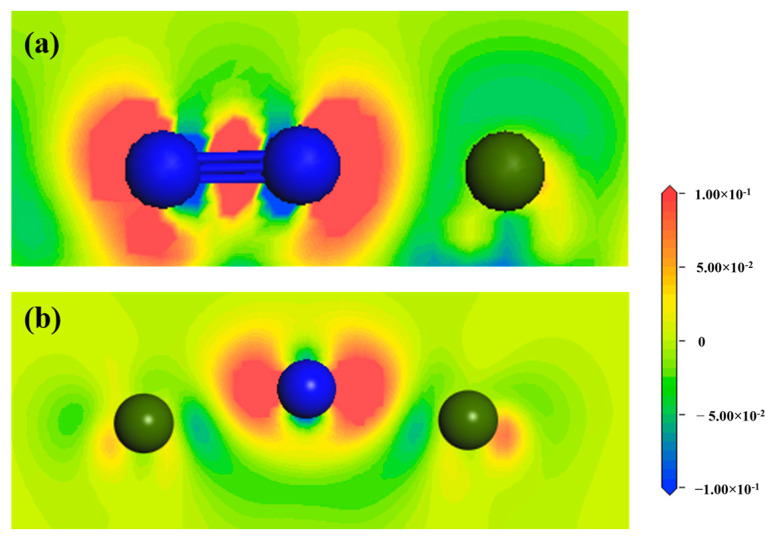
Differential charge density diagram of Hollow 1 site on the surface of ZrMnFe(110) adsorbed by (**a**) the N_2_ molecule and (**b**) N atom (the blue atom is the N atom and the green atom is the Zr atom).

**Figure 7 materials-16-03323-f007:**
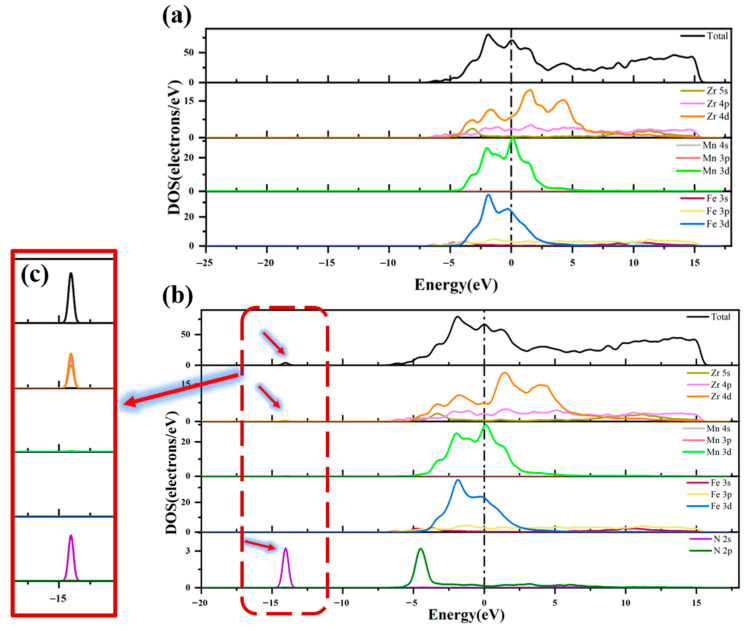
Total state density and partial state density diagram (**a**) before and (**b**) after adsorption of N atom on ZrMnFe(110) surface and (**c**) the region enlarged view.

**Figure 8 materials-16-03323-f008:**
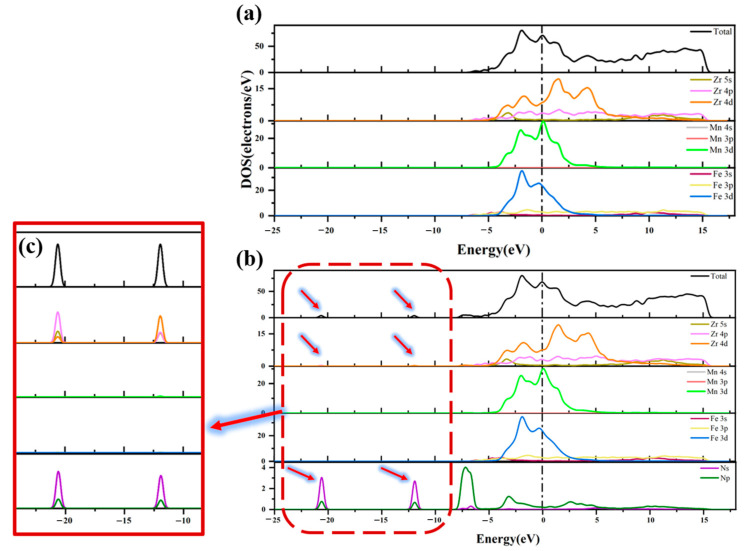
Total state density and partial state density (**a**) before and (**b**) after N_2_ adsorption on ZrMnFe(110) surface and (**c**) the region enlarged view.

**Figure 9 materials-16-03323-f009:**
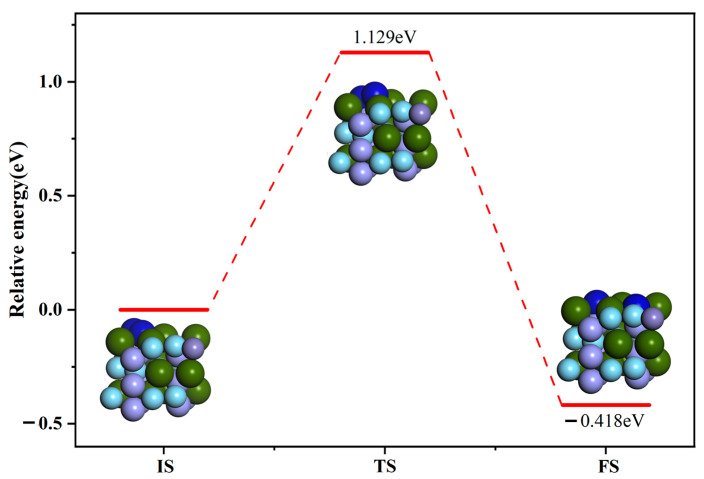
Minimum energy path diagram of N_2_ molecule dissociation on the surface of ZrMnFe(110).

**Figure 10 materials-16-03323-f010:**
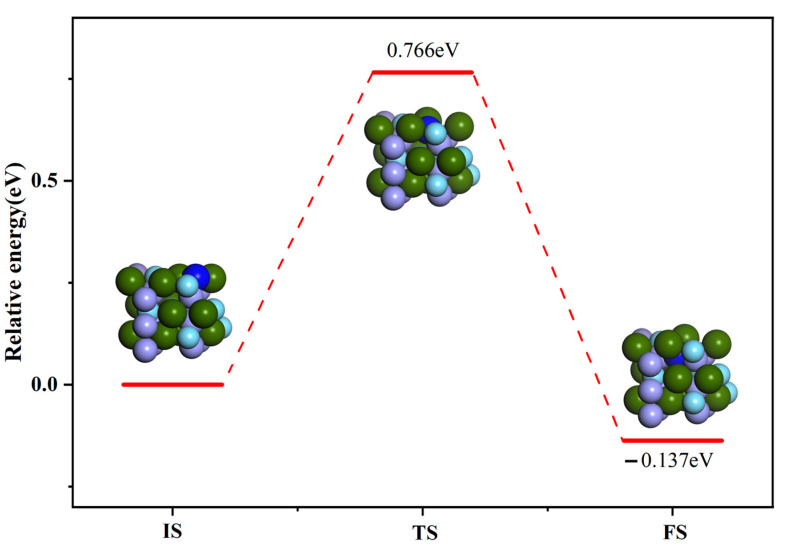
Minimum energy path of N atom on permeating ZrMnFe(110) surface.

**Table 1 materials-16-03323-t001:** Adsorption energy and adsorption position of N atom adsorbed on ZrMnFe(110) surface.

Pre-Adsorption Site	Site after Adsorption	Adsorption Energy E_a_/eV	Distance/Å
Top 1	Hollow 1	/	/
Top 2	Hollow 1	/	/
Hollow 1	Hollow 1	/	/
Hollow 2	Hollow 2	5.592	3.357
Bridge	Hollow 1	6.057	0.659

**Table 2 materials-16-03323-t002:** Adsorption energy and bond length of N_2_ adsorbed on ZrMnFe(110) surface.

Pre-Adsorption Site	Site after Adsorption	Adsorption Energy E_a_/eV	N-N Distance/Å
T1-Hor	T1-B	10.215	1.210
T2-Hor	T2-B	9.772	1.285
H1-Hor	H1	10.215	1.350
H2-Hor	H2	8.389	1.255
B-Hor	H1	10.214	1.350
T1-Ver	T1	7.807	1.127
T2-Ver	T2	8.068	1.139
H1-Ver	H1-B	9.398	1.296
H2-Ver	H2	8.389	1.238
B-Ver	H1	10.215	1.350
Free N_2_	/	/	1.099

## Data Availability

The data that support the findings of this study are available from the corresponding author, F.Z. upon reasonable request.

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
