# Peer review of "First-Principles Study of Nitrogen Adsorption and Dissociation on ZrMnFe(110) Surface"

_materials, 2023, doi:10.3390/ma16093323_

Round 1
Reviewer 1 Report
The paper from Qiaobin Yanga et al. is devoted to the theoretical investigation of nitrogen adsorption and dissociation on ZrMnFe(110) surface using first-principles calculation method
The article is quite interesting. However, the text of the article is carelessly designed. The following are the shortcomings, the elimination of which will improve the quality of the article
1. Already in the annotation there is a mention of the H1 site, although there has not been a description of what the H1 site is yet.
2. In line 25, the word "laves" apparently should be written with a capital letter "Laves"
3. Caption to Figure 1 "(a) ZrMnFe cell model and (b) ZrMnFe(110) cleaning indicator. model." apparently contains some kind of typo
4. Sentence “The adsorption of N2 molecules and N atoms on the ZrMnFe(110) surface was simulated by using pseudopotential plane wave method based on first principles in this paper [18–20], Firstly, the density mixing scheme of LBFGS (Large-Broyden -Flecher-Goldfarb-Shanno) [21-24] , combined with conjugate gradient method, was used to optimize the geometric structure of the constructed system to obtain a stable structure" should be reformulated. In this form, it looks strange.
5. In the formulas on page 3, it makes sense to introduce additional indices to distinguish between the adsorption energies of a nitrogen atom and a nitrogen molecule. For example Ea1 and Ea2.
6. On page 3, line 87, it is not entirely clear what is meant by "...the total energy of the system adsorbed on the ZrMnFe(110) surface...". Probably, the authors are talking about the total energy of the system after nitrogen adsorption on the ZrMnFe(110) surface. If something else was meant, it is necessary to explain in more detail what exactly.
7. On page 3, the authors write "As can be seen from the figure, there were only stable adsorption sites for N atoms, namely H1 and H2, while all the N atoms at T1, T2 and B sites moved to H site". However, the definition of H site has not been introduced before. Apparently there is a typo in this place
8. The caption to Figure 3, in my opinion, is somewhat incorrect, since it is not the nitrogen atom itself that is being compared, but its position on the surface after optimization.
9. The sentence "In combination with the adsorption energy of N atoms at each position on the surface of ZrMnFe(110) and the distance from the surface in Table 1, the vacancy H1 composed of 4Zr1Fe was the optimal adsorption site for N atoms on the surface of ZrMnFe(110)" is formulated unsuccessfully. In principle, it is clear what the authors meant, but it is better to reformulate this fragment of the text.
10. In the sentence "The adsorption energy for each N atom and N atom at the vacancy H2 composed of 1Zr2Fe was 6.057 eV and 5.592 eV, respectively" аn error was made. Apparently, some words are missing.
11. The sentence "It can be seen from the table that H1 was the optimal adsorption site for N2 molecule on the surface of ZrMnFe(110) and H1-hor for the horizontal adsorption for N2 molecule, with the best adsorption energy of 10.21534eV" needs to be corrected.
Firstly, the sentence itself is formulated in a strange way, perhaps some words are missing again. Secondly, the conclusion about the optimality of the H1 site simply does not follow from the table, since for horizontal adsorption the binding energies in sites H1 and T1 are equal. In addition, the initial sites are recorded in the table, not the sites after optimization. Perhaps it makes sense to show the transitions of site configurations during optimization in order to more clearly identify the most stable site. Thirdly, it is basically meaningless to give the value of the adsorption energy with an accuracy of up to the 5th decimal place. In general, the accuracy of quantum chemical calculations at the moment is such that it does not make much sense to give more than 3 significant digits for the results of such calculations. The same remark generally applies to the second column of table 2.
12. On page 6, the authors write "It can be seen from Figure 6 (a) that N2 molecule gained electrons, while Zr atom lost electrons, accompanied by a large amount of over-lapping in the electron clouds between the two N atoms, indicating the existence of N-N bond". Nevertheless, it can be seen from the figure that there are also many blue regions around the N2 molecule, which, based on the text, corresponds to a loss of electron density. It is necessary to justify the conclusion mentioned in the sentence more carefully.
13. Figure 6 is poorly designed. The atoms in the drawing are not signed. The interaction of a nitrogen molecule is shown with one zirconium atom, and the interaction of a nitrogen atom with two zirconium atoms. It is not clear why the conclusion is made that the nitrogen molecule acquires a charge, and the zirconium atom loses?
14. Figures 7 and 8 are poorly designed. In the captions to the figures, it is desirable to specify in detail what exactly is depicted on each part of the figure (a, b, c). The images on parts of figures a and b, as well as the insertion on part c, are too small. The scales are simply not visible. Images need to be enlarged.
15. Paragraph "Figure 7 compares the DOS diagrams before and after the adsorption of N2 molecule at H1 site on the ZrMnFe(110) surface. As can be seen from the figure, the shape of the total state density diagram basically remained unchanged after adsorption but the total state density of the adsorption surface appeared new peaks at -20.61 eV and -20.61 eV, respectively. It can be seen from PDOS that these two peaks were contributed by the state density peak of N atom, indicating that the state density of N2 molecule were split after adsorption. Hence, it can be concluded that N2 molecule had been dissociated to a certain extent. In addition, as shown in Figure 7 (c), new density peaks appeared at -20.61 eV and -20.61 eV in the 4p orbitals of Zr after adsorption and adsorption, respectively, which was also attributed to the hybridization of N atom by Zr atomic orbital" needs to be carefully reworked, it has too many errors.
Firstly, this paragraph probably refers to Figure 8, not Figure 7. Secondly, two DOS peaks are mentioned several times, but the energy for both peaks is the same (20.61 eV). Thirdly, the phrase "...after adsorption and adsorption, respectively..." clearly contains a typo. In addition, the term "adsorption surface" is not quite clear, as well as the somewhat previously used term "adsorbed surface".
16. On page 7, the authors write "Dissociation chemisorption is a key step in most surface chemistry. Therefore, in order to study the nitridation mechanism of ZrMnFe(110) surface, the adsorption and desilption characteristics of N2 molecules should be studied in detail". Perhaps dissociative chemisorption was meant? The term "desilption" is also unclear.
17. It is not entirely clear what the authors meant in the sentence "With the occurrence of dissociation reaction, N2 molecule gradually tilted and N atom on the upper side gradually moved to other vacancies on the ZrMnFe(110) surface until it moves to the vacancy"
Most of these shortcomings relate to the design of the article and the general style of the text. Perhaps the authors should consult with a native English speaker.
After correcting these shortcomings, the article can be accepted for publication
Reviewer 2 Report
Article discuss basic problems on gas sorption onto surfaces and the obtained conclusions deserves publication. High class modelling etc approach is used to develop new understanding on nitrogen interaction with surface located metal atoms. The conclusions are argumented and supported by the obtained results.
Still some corrections should be done:
1. Writing of chemical formulas should be corrected: writing of formulas such as N2, H2O, CH4 is unacceptable
2. Fig 6 - legend should be provided to color scale
3. Fig. 7 - too small and not readable
4. Author contribution at the end should be indicated
5. List of references is a mess: it should be formatised accordingly requirements
